# MODEL COMPRESSION VIA HYPER-STRUCTURE NETWORK

## ABSTRACT

In this paper, we propose a novel channel pruning method to solve the problem of compression and acceleration of Convolutional Neural Networks (CNNs). Previous channel pruning methods usually ignore the relationships between channels and layers. Many of them parameterize each channel independently by using gates or similar concepts. To fill this gap, a hyper-structure network is proposed to generate the architecture of the main network. Like the existing hypernet, our hyper-structure network can be optimized by regular backpropagation. Moreover, we use a regularization term to specify the computational resource of the compact network. Usually, FLOPs is used as the criterion of computational resource. However, if FLOPs is used in the regularization, it may over penalize early layers. To address this issue, we further introduce learnable layer-wise scaling factors to balance the gradients from different terms, and they can be optimized by hyper-gradient descent. Extensive experimental results on CIFAR-10 and ImageNet show that our method is competitive with state-of-the-art methods.

## 1 INTRODUCTION

Convolutional Neural Networks (CNNs) have accomplished great success in many machine learning and computer vision tasks (Krizhevsky et al., 2012; Redmon et al., 2016; Ren et al., 2015; Simonyan & Zisserman, 2014a; Bojarski et al., 2016). To deal with real world applications, recently, the design of CNNs becomes more and more complicated in terms of width, depth, etc. (Krizhevsky et al., 2012; Simonyan & Zisserman, 2014b; He et al., 2016; Huang et al., 2017). Although these complex CNNs can attain better performance on benchmark tasks, their computational and storage costs increase dramatically. As a result, a typical application based on CNNs can easily exhaust an embedded or mobile device due to its enormous costs. Given such costs, the application can hardly be deployed on resource-limited platforms. To tackle these problems, many methods (Han et al., 2015b;a) have been devoted to compressing the original large CNNs into compact models. Among these methods, weight pruning and structural pruning are two popular directions.

Unlike weight pruning or sparsification, structural pruning, especially channel pruning, is an effective way to truncate the computational cost of a model because it does not require any post-processing steps to achieve actual acceleration and compression. Many existing works (Liu et al., 2017; Ye et al., 2018; Huang & Wang, 2018; Kim et al., 2020; You et al., 2019) try to solve the problem of structure pruning by applying gates or similar concepts on channels of a layer. Although these ideas have achieved many successes in channel pruning, there are some potential problems. Usually, each gate has its own parameter, but parameters from different gates do not have dependence. As a result, they can hardly learn inter-channel or inter-layer relationships. Due to the same reason, the slimmed models from these methods could overlook the information between different channels and layers, potentially bringing sub-optimal model compression results.

To address these challenges, we propose a novel channel pruning method inspired by hypernet (Ha et al., 2016). In hypernet, they propose to use a hyper network to generate the weights for another network, while the hypernet can be optimized through backpropagation. We extend a hypernet to a hyper-structure network to generate an architecture vector for a CNN instead of weights. Each architecture vector corresponds to a sub-network from the main (original) network. By doing so, the inter-channel and inter-layer relationships can be captured by our hyper-structure network.

Besides the hyper-structure network, we also introduce a regularization term to control the computational budget of a sub-network. Recent model compression methods focus on pruning computational FLOPs instead of parameters. The problem of applying FLOPs regularization is that the gradients of the regularization will heavily penalize early layers which can be regarded as a bias towards latter layers. Such a bias will restrict the potential search space of sub-networks. To make our hyper-structure network explore more possible structures, we further introduce layer-wise scaling factors to balance the gradients from different losses for each layer. These factors can be optimized by hyper-gradient descent.

Our contributions are summarized as follows:

1) Inspired by hypernet, we propose to use a hyper-structure network for model compression to capture inter-channel and inter-layer relationships. Similar to hypernet, the proposed hyper-structure network can be optimized by regular backpropagation.
2) Gradients from FLOPs regularization are biased toward latter layers, which truncate the potential search space of a sub-network. To balance the gradients from different terms, layer-wise scaling factors are introduced for each layer. These scaling factors can be optimized through hyper-gradient descent with trivial additional costs.
3) Extensive experiments on CIFAR-10 and ImageNet show that our method can outperform both conventional channel pruning methods and AutoML based pruning methods on ResNet and MobileNetV2.

## 2 RELATED WORKS

### 2.1 MODEL COMPRESSION

Recently, model compression has drawn a lot of attention from the community. Among all model compression methods, weight pruning and structural pruning are two popular directions.

Weight pruning eliminates redundant connections without assumptions on the structures of weights. Weight pruning methods can achieve a very high compression rate while they need specially designed sparse matrix libraries to achieve acceleration and compression. As one of the early works, Han et al. (2015b) proposes to use $L_1$ or $L_2$ magnitude as the criterion to prune weights and connections. SNIP (Lee et al., 2019) updates the importance of each weight by using gradients from loss function. Weights with lower importance will be pruned. Lottery ticket hypothesis (Frankle & Carbin, 2019) assumes there exist high-performance sub-networks within the large network at initialization time. They then retrain the sub-network with the same initialization. In rethinking network pruning (Liu et al., 2019b), they challenging the typical model compression process (training, pruning, fine-tuning), and argue that fine-tuning is not necessary. Instead, they show that training the compressed model from scratch with random initialization can obtain better results.

One of the previous works (Li et al., 2017) in structural pruning uses the sum of the absolute value of kernel weights as the criterion for filter pruning. Instead of directly pruning filters based on magnitude, structural sparsity learning (Wen et al., 2016) is proposed to prune redundant structures with Group Lasso regularization. On top of structural sparsity, GrOWL regularization is applied to make similar structures share the same weights (Zhang et al., 2018). One of the problems when using Group Lasso is that weights with small values could still be important, and it's difficult for structures under Group Lasso regularization to achieve exact zero values. As a result, Louizos et al. (2018) propose to use explicit $L_0$ regularization to make weights within structures have exact zero values. Besides using the magnitude of structure weights as a criterion, other methods utilize the scaling factor of batchnorm to achieve structure pruning, since batchnorm (Ioffe & Szegedy, 2015) is widely used in recent neural network designs (He et al., 2016; Huang et al., 2017). A straightforward way to achieve channel pruning is to make the scaling factor of batchnorm to be sparse (Liu et al., 2017). If the scaling factor of a channel fell below a certain threshold, then the channel will be removed. The scaling factor can also be regarded as the gate parameter of a channel. Methods related to this concept include (Ye et al., 2018; Huang & Wang, 2018; Kim et al., 2020; You et al., 2019). Though it has achieved many successes in channel pruning, using gates can not capture the relationships between channels and across layers. Besides using gates, Collaborative channel pruning (Peng et al., 2019) try to prune channels by using Taylor expansion. Our method is also related to Automatic Model Compression(AMC) (He et al., 2018b). In AMC, they use policy gradient to update the policy

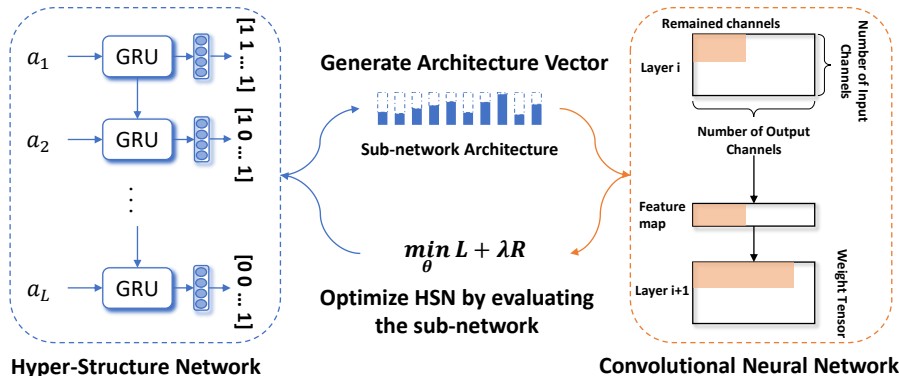

Figure 1: Overview of our proposed method. The width and height dimension of weight tensors are omitted. The architecture vector $\mathbf{v}$ is firstly generated from fixed input $a_i$, $i = 1, \ldots, L$. Then, a sub-network is sampled according to the architecture vector $\mathbf{v}$. The parameters of HSN are updated by using gradients from the loss function when evaluating the sub-network.

network, which potentially provides both inter-channel and inter-layer information. However, the high variance of policy gradient makes it less efficient and effective compared to our method. In this paper, we focus on channel pruning, since it provides a natural way to reduce computation and memory costs.

Besides weight and channel pruning methods, there are works from other perspectives, including bayesian pruning (Molchanov et al., 2017; Neklyudov et al., 2017), weight quantization (Courbariaux et al., 2015; Rastegari et al., 2016), and knowledge distillation (Hinton et al., 2015).

### 2.2 HYPERNET

Hypernet (Ha et al., 2016) was introduced to generate weights for a network by using a hyper network. Hyper networks have been applied to many machine learning tasks. von Oswald et al. (2020) uses a hyper network to generate weights based on task identity to combat catastrophic forgetting in continual learning. MetaPruning (Liu et al., 2019a) utilizes a hyper network to generate weights when performing evolutionary algorithm. SMASH (Brock et al., 2018) is a neural architecture search method that can predict the weights of a network given its architecture. GPN (Zhang et al., 2019) extends the idea of SMASH and can be used on any directed acyclic graph. Other applications include Bayesian neural networks (Krueger et al., 2017), multi-task learning (Pan et al., 2018), generative models (Suarez, 2017) and so on. Different from original hyper network, the proposed hyper-structure network aims to generate the architecture of a sub-network.

## 3 PROPOSED METHOD

### 3.1 NOTATIONS

To better describe our proposed approach, necessary notations are introduced first. In a CNN, the feature map of $i$th layer can be represented by $\mathcal{F}_i \in \Re^{C_i \times W_i \times H_i}$, $i = 1, \ldots, L$, where $C_i$ is the number of channels, $W_i$ and $H_i$ are height and width of the current feature map, $L$ is the number of layers. The mini-batch dimension of feature maps is ignored to simplify notations. sigmoid($\cdot$) is the sigmoid function. round($\cdot$) rounds inputs to nearest integers.

### 3.2 HYPER-STRUCTURE NETWORK

In the context of channel pruning, we need to decide whether a channel should be pruned or not. We can use 0 or 1 to depict the removal or keep of a channel. Consequently, the architecture of a sub-network can be represented as a concatenated vector (containing 0 or 1) from all layers. Our goal is then to use a neural network to generate this vector to represent the corresponding sub-network.

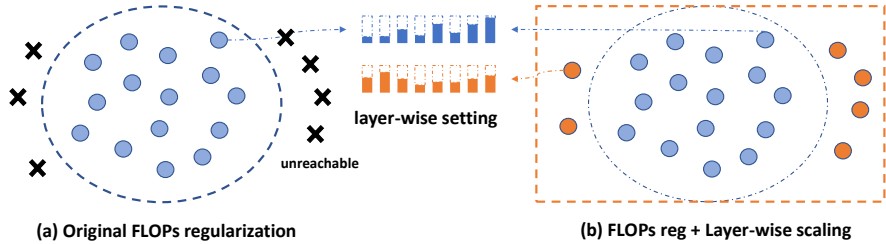

(a) Original FLOPs regularization               (b) FLOPs reg + Layer-wise scaling

Figure 2: **(a)** For the original FLOPs regularization, some architectures may become unreachable. **(b)** After layer-wise scaling, the potential search space of architectures for a sub-network is increased.

For $i$th layer, the following output vector is generated:

$$o_i = \text{HSN}(a_i; \Theta), \tag{1}$$

where HSN is our proposed hyper-structure network composed of gated recurrent unit (GRU) (Cho et al., 2014) and dense layers, $a_i$ is a fixed random vector generated from a uniform distribution $\mathcal{U}(0, 1)$, and $\Theta$ is the parameter of HSN. The detailed setup of HSN can be found in Appendix C. In short, GRU is used to capture sequential relationships between layers, and dense layers are capable of capturing inter-channel relationships. Note that $a_i$ is a constant vector during training, if $a_i$ is randomly sampled, it will make learning more difficult and result in sub-optimal performance.

Now we have the output $o_i$, we need to convert it to a 0-1 vector to evaluate the sub-network. The binarization process can be demonstrated by the following equations:

$$\begin{aligned} z_i &= \text{sigmoid}((o_i + g)/\tau), \\ v_i &= \text{round}(z_i), \text{ and } v_i \in \{0, 1\}^{C_i}, \end{aligned} \tag{2}$$

where $g$ follows Gumbel distribution: $g \sim \text{Gumbel}(0, 1)$, $v_i$ is the architecture vector of $i$th layer, and $\tau$ is the temperature hyper-parameter. Since the round operation is not differentiable, we use straight through estimator (STE) (Bengio et al., 2013) to enable gradient calculation: $\frac{\partial \mathcal{J}}{\partial z_i} = \frac{\partial \mathcal{J}}{\partial v_i}$. This process can be summarized as using ST Gumbel-Softmax (Jang et al., 2016) with fixed temperature to approximate Bernoulli distribution. The idea of HSN can also be viewed as mapping from constant vectors $\{a_i\}_{i=1}^{L}$ to the architecture of a sub-network. When we evaluate a sub-network, the feature map of $i$th layer is modified as follows:

$$\widehat{\mathcal{F}}_i = \hat{v}_i \odot \mathcal{F}_i, \tag{3}$$

where $\odot$ is element-wise multiplication, $\hat{v}_i$ is the expanded version of $v_i$, and $\hat{v}_i$ has the same size of $\mathcal{F}_i$. The feature map $\mathcal{F}_i$ is from the output of Conv-Bn-Relu block. The overall loss function is:

$$\min_{\Theta} \mathcal{J}(\Theta) := \mathcal{L}\big(f(x; \mathcal{W}, \mathbf{v}), y\big) + \lambda \mathcal{R}(T(\mathbf{v}), pT_{\text{total}}) \tag{4}$$

where $\mathbf{v} = (v_1, \dots, v_L)$, $T(\mathbf{v})$ is the current FLOPs decided by the architecture vector $\mathbf{v}$, $T_{\text{total}}$ is the total FLOPs of the original model, $p \in (0, 1]$ is a predefined parameter deciding the remaining fraction of FLOPs, $\lambda$ is the hyper-parameter controlling the strength of FLOPs regularization, $f(x; \mathcal{W}, \mathbf{v})$ is the CNN parameterized by $\mathcal{W}$ and the sub-network structure is determined by architecture vector $\mathbf{v}$, $\mathcal{L}$ is the cross entropy loss function and $\mathcal{R}$ is the regularization term for FLOPs, and $\Theta$ again is the parameters of HSN. The regularization term $\mathcal{R}$ used in this paper is $\mathcal{R}(T(\mathbf{v}), pT_{\text{total}}) = \log(|T(\mathbf{v}) - pT_{\text{total}}| + 1)$.

### 3.3 LAYER-WISE SCALING

The FLOPs regularization considered in Eq. 4 will heavily penalize layers with a larger amount of FLOPs (early layers for most architectures). Consequently, the resulting architecture from the original FLOPs regularization will have a larger pruning rate at early layers. The alternative architectures with similar FLOPs could be omitted. This phenomenon is also demonstrated in Fig. 2. Further analysis is provided in Appendix B.

To alleviate the problem caused by original FLOPs regularization, we introduce layer-wise scaling factors to dynamically balance gradients from the regularization term $\mathcal{R}$ and the loss term $\mathcal{L}$. Only

---

**Algorithm 1:** Model Compression via Hyper-Structure Network

---

**Input**: dataset for training HSN: $D_{\text{HSN}}$; perversed rate of FLOPs: $p$; hyper-parameter: $\lambda$; training epochs: $n_E$; pre-trained CNN: $f$; learning rate $\beta$ when updating $\{\alpha_i\}_{i=1}^L$.
**Initialization**: initialize $\Theta$ randomly; initialize $\alpha_i = 1$, $i = 1, \ldots, L$; freeze $\mathcal{W}$ in $f$.
**for** $e := 1$ *to* $n_E$ **do**
$\quad$ shuffle($D_{\text{HSN}}$)
$\quad$ **for** *a batch* $(x, y)$ *in* $D_{HSN}$ **do**
$\quad\quad$ 1. produce architecture vector $\mathbf{v}$ from HSN (Eq. 1 and 2)
$\quad\quad$ 2. calculate gradients w.r.t $\Theta$ (Eq. 5).
$\quad\quad$ 3. calculate hyper-gradient for $\alpha_i$ (Eq. 6).
$\quad\quad$ 4. update layer-wise scaling factor $\alpha_i^t = \alpha_i^{t-1} - \beta \frac{\partial J(u(\theta_i^{t-2}, \alpha_i^{t-1}))}{\partial \alpha_i}, i = 1, \ldots, L$.
$\quad\quad$ 5. update $\Theta$ by ADAM optimizer.
$\quad$ **end**
**end**
**return** HSN with the final $\Theta$.

---

gradients in dense layers are balanced since GRU is shared by all layers. The gradients w.r.t the parameters of $i$th dense layer can be written in the following equation:

$$\frac{\partial \mathcal{J}}{\partial \theta_i} = \alpha_i \frac{\partial \mathcal{L}}{\partial \theta_i} + \lambda \frac{\partial \mathcal{R}}{\partial \theta_i}, \tag{5}$$

where $\theta_i$ is the parameter of $i$th dense layer, $\alpha_i$ is the layer-wise scaling factor for $i$th layer. If no layer-wise scaling is applied, $\alpha_i = 1$. $\alpha_i$ can be regarded as a balancing factor between $\frac{\partial \mathcal{R}}{\partial \theta_i}$ and $\frac{\partial \mathcal{L}}{\partial \theta_i}$. $\alpha_i$ only appears in gradient calculation, as a result, it can not be directly optimized. To optimize $\alpha_i$, we follow similar deriving process from (Baydin et al., 2018). We first define the update rule $\theta_i^t = u(\theta_i^{t-1}, \alpha_i^t)$ and it can be applied to any optimization algorithms. For example, under stochastic gradient descent, $u(\theta_i^{t-1}, \alpha_i^t) = \theta_i^{t-1} - \eta(\alpha_i^t \frac{\partial \mathcal{L}}{\partial \theta_i^{t-1}} + \lambda \frac{\partial \mathcal{R}}{\partial \theta_i^{t-1}})$. Ideally, our goal is to update $\alpha_i$ so that the corresponding architecture can obtain lower loss value with loss function $\mathcal{J}$. To do so, we want to $\min_{\alpha_i} J(u(\theta_i^{t-1}, \alpha_i^t))$ before update $\theta_i$. For simplicity, the expectation is omitted. The hyper-gradient with respect to $\alpha_i$ can be calculated by:

$$\frac{\partial J(u(\theta_i^{t-1}, \alpha_i^t))}{\partial \alpha_i} = (\frac{\partial \mathcal{J}}{\partial u})^T \frac{\partial u(\theta_i^{t-1}, \alpha_i^t)}{\partial \alpha_i} = (\frac{\partial \mathcal{J}}{\partial \theta_i^t})^T \frac{\partial u(\theta_i^{t-1}, \alpha_i^t)}{\partial \alpha_i}. \tag{6}$$

Given the hyper-gradient of $\alpha_i$, it can be updated by regular gradient descent method. In experiments, the update rule is ADAM optimizer (Kingma & Ba, 2014), and the detail derivation of $\frac{\partial J(u(\theta_i^{t-1}, \alpha_i^t))}{\partial \alpha_i}$ for ADAM optimizer is described in Appendix E.

### 3.4 Model Compression via Hyper-Structure Network

In Fig. 6, we provide the flowchart of HSN. The overall algorithm of model compression via hyper-structure network is shown in Alg. 1. As shown in Alg. 1, our method can prune any pre-trained CNNs without modifications. It should be emphasized again that the gradient of GRU is not affected by $\alpha_i$, which is simply $\frac{\partial \mathcal{L}}{\partial \theta_{\text{GRU}}} + \lambda \frac{\partial \mathcal{R}}{\partial \theta_{\text{GRU}}}$, and $\theta_{\text{GRU}}$ is the parameter for GRU. Moreover, HSN does not need a whole dataset for training, and a small fraction of the dataset is enough, which makes the training of HSN quite efficient. After the training of HSN, we then use HSN to generate an architecture vector $\mathbf{v}$, and prune the model according to this vector. Also, note that there is certain randomness (Eq. 2 approximates Bernoulli distribution) when generating $\mathbf{v}$, but we find that there is no need to generate the vector multiple times, and average them or conduct majority vote. When generating the vector multiple times, most parts of vectors are the same, the different parts are trivial and do not have impacts on the final performance.

| Method | Architecture | Baseline Acc | Pruned Acc | $\Delta$-Acc | $\downarrow$ FLOPs |
|---|---|---|---|---|---|
| Channel Pruning (He et al., 2017) | | 92.80% | 91.80% | -1.00% | 50.0% |
| AMC (He et al., 2018b) | | 92.80% | 91.90% | -0.90% | 50.0% |
| Pruning Filters (Li et al., 2017) | | 93.04% | 93.06% | +0.02% | 27.6% |
| Soft Prunings (He et al., 2018a) | ResNet-56 | 93.59% | 93.35% | -0.24% | 52.6% |
| DCP (Zhuang et al., 2018) | | 93.80% | 93.59% | -0.31% | 50.0% |
| DCP-Adapt (Zhuang et al., 2018) | | 93.80% | 93.81% | +0.01% | 47.0% |
| CCP (Peng et al., 2019) | | 93.50% | 93.42% | -0.08% | **52.6%** |
| MCH(ours) | | 92.99% | 93.23% | **+0.24%** | 50.0% |
| WM (Zhuang et al., 2018) | | 94.47% | 94.17% | -0.30% | 26.0% |
| DCP (Zhuang et al., 2018) | MobileNetV2 | 94.47% | 94.69% | +0.22% | 26.0% |
| MCH(ours) | | 94.23% | 94.68% | **+0.38%** | **40.0%** |

Table 1: Comparison results on CIFAR-10 dataset with ResNet-56 and MobileNetV2. $\Delta$-Acc represents the performance changes before and after model pruning. +/- indicates increase or decrease compared to baseline results.

## 4 EXPERIMENTAL RESULTS

### 4.1 IMPLEMENTATION DETAILS

Similar to many model compression works, CIFAR-10 (Krizhevsky & Hinton, 2009) and ImageNet (Deng et al., 2009) are used to evaluate the performance of our method. Our method requires one hyper-parameter $p$ to control the FLOPs budget. The detailed choices of $p$ are listed in Appendix F.

For CIFAR-10, we compare with other methods on ResNet-56 and MobileNetV2. For ImageNet, we select ResNet-34, ResNet-50, ResNet-101 and MobileNetV2 as our target models. The reason we choose these models is because that ResNet (He et al., 2016) and MobileNetV2 (Sandler et al., 2018) are much harder to prune than earlier models like AlexNet (Krizhevsky et al., 2012) and VGG (Simonyan & Zisserman, 2014b). $\lambda$ decides the regularization strength in our method. We choose $\lambda = 4$ in all CIFAR-10 experiments and $\lambda = 8$ for all ImageNet experiments.

For CIFAR-10 models, we train ResNet-56 from scratch following the pytorch examples. After pruning, we finetune the model for 160 epochs using SGD with a start learning rate 0.1, weight decay 0.0001 and momentum 0.8, the learning rate is multiplied by 0.1 at epoch 80 and 120. For ImageNet models, we directly use the pre-trained models released from pytorch (Paszke et al., 2017; 2019). After pruning, we finetune the model for 100 epochs using SGD with a start learning rate 0.01, weight decay 0.0001 and momentum 0.9, and the learning rate is scaled by 0.1 at epoch 30, 60 and 90. For MobileNetV2 on ImageNet, we choose weight decay as 0.00004 which is the same with the original paper (Sandler et al., 2018).

For the training process of HSN, we use ADAM (Kingma & Ba, 2014) optimizer with a constant learning rate 0.001 and train HSN for 200 epochs. $\tau$ in Eq. 2 is set as 0.4. The $\beta$ for $\alpha_i$ is chosen as 0.01, and $\alpha_i$ is updated as shown in Alg. 1. To build dataset $D_{\text{HSN}}$, we random sample $2,500$ and $10,000$ samples for for CIFAR-10 and ImageNet separately. In the experiments, we found that a stand-alone validation set is not necessary, all samples in $D_{\text{HSN}}$ come from the original training set. All codes in this paper are implemented with pytorch (Paszke et al., 2017; 2019). The experiments are conducted on a machine with 4 Nvidia Tesla P40 GPUs.

### 4.2 CIFAR-10 RESULTS

In Tab. 1, we present the comparison results on CIFAR-10 dataset. Our method is abbreviated as MCH (**M**odel **C**ompression via **H**yper Structure Network) in the experiment section. For ResNet-56, our method can prune $50\%$ of FLOPs while obtain $0.24\%$ performance gain in accuracy. On MobileNetV2, our method can obtain $0.38\%$ gain in accuracy. Compared to all other methods, our method can achieve the best results. Our method can outperform the second best method (DCP-Adapt) by $0.23\%$ on ResNet-56. On MobileNetV2, our method can outperform the the second best method by $0.16\%$ while pruning $14\%$ more FLOPs. For both models, our method performs much better than early methods (He et al., 2017; 2018b; Li et al., 2017; He et al., 2018a). Our method can outperform

| Method | Architecture | Pruned Top-1 | Pruned Top-5 | Δ Top-1 | Δ Top-5 | ↓ FLOPs |
|---|---|---|---|---|---|---|
| Pruning Filters (Li et al., 2017) | | 72.17% | - | -1.06% | - | 24.8% |
| Soft Prunings (He et al., 2018a) | | 71.84% | 89.70% | -2.09% | -1.92% | 41.1% |
| IE (Molchanov et al., 2019) | ResNet-34 | 72.83% | - | -0.48% | - | 24.2% |
| FPGM (He et al., 2019) | | 72.63% | 91.08% | -1.29% | -0.54% | 41.1% |
| MCH(ours) | | **72.85%** | **91.15%** | **-0.45%** | **-0.27%** | **44.0%** |
| IE (Molchanov et al., 2019) | | 74.50% | - | -1.68% | - | 45.0% |
| FPGM (He et al., 2019) | | 74.83% | 92.32% | -1.32% | -0.55% | 53.5% |
| GAL (Lin et al., 2019) | | 71.80% | 90.82% | -4.35% | -2.05% | 55.0% |
| DCP (Zhuang et al., 2018) | | 74.95% | 92.32% | -1.06% | -0.61% | 55.6% |
| CCP (Peng et al., 2019) | ResNet-50 | 75.21% | 92.42% | -0.94% | -0.45% | 54.1% |
| MetaPruning (Liu et al., 2019a) | | 75.40% | - | -1.20% | - | 51.2% |
| GBN (You et al., 2019) | | 75.18% | 92.41% | -0.67% | -0.26% | 55.1% |
| HRank (Lin et al., 2020) | | 74.98% | 92.33% | -1.17% | -0.54% | 43.8% |
| Hinge (Li et al., 2020) | | 74.70% | - | -1.40% | - | 54.4% |
| LeGR (Chin et al., 2020) | | 75.30% | - | -0.80% | - | 54.0% |
| MCH(ours) | | **75.60%** | **92.67%** | **-0.55%** | **-0.20%** | **56.0%** |
| Rethinking (Ye et al., 2018) | | 77.37% | - | -2.10% | - | 47.0% |
| IE (Molchanov et al., 2019) | ResNet-101 | 77.35% | - | -0.02% | - | 39.8% |
| FPGM (He et al., 2019) | | 77.32% | 93.56% | -0.05% | 0.00% | 41.1% |
| MCH(ours) | | **77.58%** | **93.81%** | **+0.21%** | **+0.25%** | **56.0%** |
| MobileNetV2 0.75 (Sandler et al., 2018) | | 69.80% | 89.60% | -2.00% | -1.40% | 30.0% |
| AMC (He et al., 2018b) | | 70.80% | - | -1.00% | - | 30.0% |
| MetaPruning (Liu et al., 2019a) | MobileNetV2 | 71.20% | - | -0.80% | - | **30.7%** |
| LeGR (Chin et al., 2020) | | 71.40% | - | -0.40% | - | 30.0% |
| MCH(ours) | | 71.54% | 90.08% | -0.58% | -0.33% | 30.1% |
| MCH(cos scheduler) | | **71.73%** | **90.17%** | **-0.39%** | **-0.24%** | 30.1% |

Table 2: Comparison results on ImageNet dataset with ResNet-34, ResNet-50, ResNet-101 and MobileNetV2. Δ-Acc represents the performance changes before and after model pruning. +/- indicates increase or decrease compared to baseline results.

CCP by $0.32\%$ in terms of Δ-Acc, which demonstrate that learning both inter-layer and inter-channel relationships are better than only considering inter-channel relationships.

## 4.3 IMAGENET RESULTS

In Tab. 2, the results on ImageNet are presented, Top-1/Top-5 accuracy after pruning are presented. Most of our comparison methods comes from recently published papers including IE (Molchanov et al., 2019), FPGM (He et al., 2019), GAL (Lin et al., 2019), CCP (Peng et al., 2019), MetaPruning (Liu et al., 2019a), GBN (You et al., 2019), Hinge (Li et al., 2020), abd HRank (Lin et al., 2020).

**ResNet-34**. Our proposed MCH can prune $44.0\%$ of FLOPs with $0.45\%$ and $0.27\%$ performance loss on Top-1 and Top-5 accuracy. Such a result is better than any other method. Proposed MCH performs similarly compared to IE (Molchanov et al., 2019) in Top-1 and Δ Top-1 accuracy ($72.85\%/-0.45\%$ vs. $72.83\%/-0.43\%$), while our method can prune almost $20\%$ more FLOPs. Given similar FLOPs pruning rate, our method achieves better results compared to FPGM (He et al., 2019) ($-0.45\%/-0.27\%$ vs. $-1.29\%/-0.54\%$ for Δ Top-1/Δ Top-5). Besides IE and FPGM, The margin between our method and rest methods are even larger.

**ResNet-50**. ResNet-50 is a very popular model for evaluating model compression methods. With such intense competition, our method can still achieve the best Top-1/Top-5 and Δ Top-1/Δ Top-5 results. The second best method in terms of Top-1 accuracy is MetaPruning (Liu et al., 2019a), which can achieve $75.40\%$ Top-1 result after pruning. Our method outperforms MetaPruning by $0.20\%$ in Top-1 accuracy while our method can prune $5\%$ more FLOPs. MetaPruning utilizes hypernet to generate weights when evaluating sub-networks, however, such a design paradigm prohibits MetaPruning to be directly used on pre-trained models. The weights inherited from the pre-trained model might be one of the reasons why our method can outperform MetaPruning. GBN (You et al., 2019) obtains the second-best Δ Top-1 accuracy, however, the accuracy after pruning is quite low compared to other methods. Our method can outperform GBN by $0.42\%$ in Top-1 accuracy. Besides GBN and MetaPruning, our method can outperform two recent methods HRank (Lin et al., 2020) and Hinge (Li et al., 2020) by $0.62\%$ to $0.90\%$ on Top-1 accuracy.

**ResNet-101**. For ResNet-101, our method can increase the performance of the baseline model by $0.21\%$ and $0.25\%$ on Top-1 and Top-5 accuracy, while removes $56\%$ of FLOPs. The second best

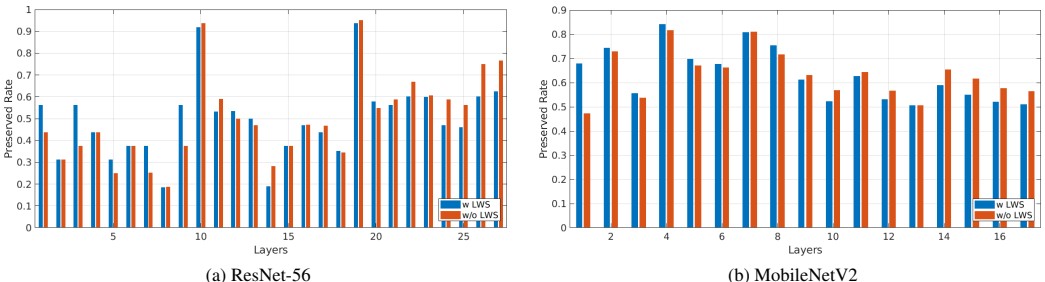

Figure 3: (a,b): Effect of $\lambda$ on the performance of sub-networks. (c,d): Effect of layer-wise scaling on the performance of sub-network. All experiments are done on CIFAR-10.

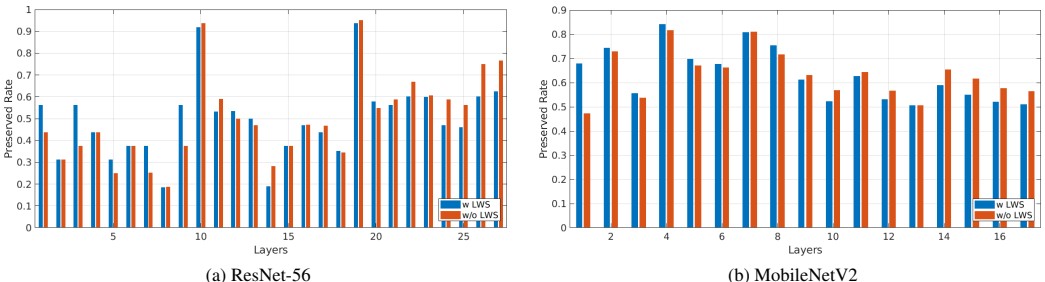

Figure 4: Layer-wise preserved rate with or without layer-wise scaling (LWS) for ResNet-56 and MobileNetV2 on CIFAR-10.

method FPGM (He et al., 2019) can maintain the performance and reducing $41\%$ of FLOPs. In short, compared to FPGM, our method can obtain performance gain while pruning $15\%$ more FLOPs.

**MobileNetV2**. On MobileNetV2, we mainly compare with AMC (He et al., 2018b) and MetaPruning (Liu et al., 2019a). Both of them can be regarded as representative works for AutoML related model compression methods (AMC uses reinforcement learning; MetaPruning uses evolutionary algorithm and hypernet). Our method can achieve $71.54\%$ Top-1 accuracy while pruning around $30\%$ of FLOPs, which is $0.34\%$ and $0.74\%$ higher than MetaPruning and AMC. These results show that our method can outperform AutoML based methods.

In summary, our method can outperform these comparison methods and achieve the state-of-the-art performance. These experimental results also indicate that inter-channel and inter-layer relationships should be considered when designing model compression methods.

### 4.4 EFFECTS OF LAYER-WISE SCALING

We further study the impact of $\lambda$ and layer-wise scaling (LWS) when training HSN on CIFAR-10. In Fig. 3 (a,b), we can see that changing $\lambda$ does not have a large impact on the final performance of a sub-network, and our method is not sensitive to it. One possible reason is that $\alpha_i$ adapts to $\lambda$ when using ADAM optimizer. In general, we do not spend too much time on tuning $\lambda$. In Fig. 3 (c,d), it shows that using LWS can improve the final performance of a sub-network and obtain lower loss. Moreover, early layers usually have a larger preserved rate with LWS as shown in Fig 4, indicating that alternative sub-network architectures can be discovered from LWS. Without LWS, the final performance of ResNet-56 will decrease $0.19\%$, achieves $93.04\%$ final accuracy on CIFAR-10. Similar observations hold for MobileNetV2 ($94.45\%$ final accuracy and the relative gap is $0.16\%$). These observations show that LWS indeed helps the training of the HSN.

### 4.5 DETAILED ANALYSIS

In this section, we provide detailed analysis to answer the following questions: **(1)** why we use fixed inputs for $a_i$? **(2)** Can we replace HSN with dense layers? **(3)** Does LWS work for different learning rate settings? **(4)** Does LWS still work for other optimization methods?

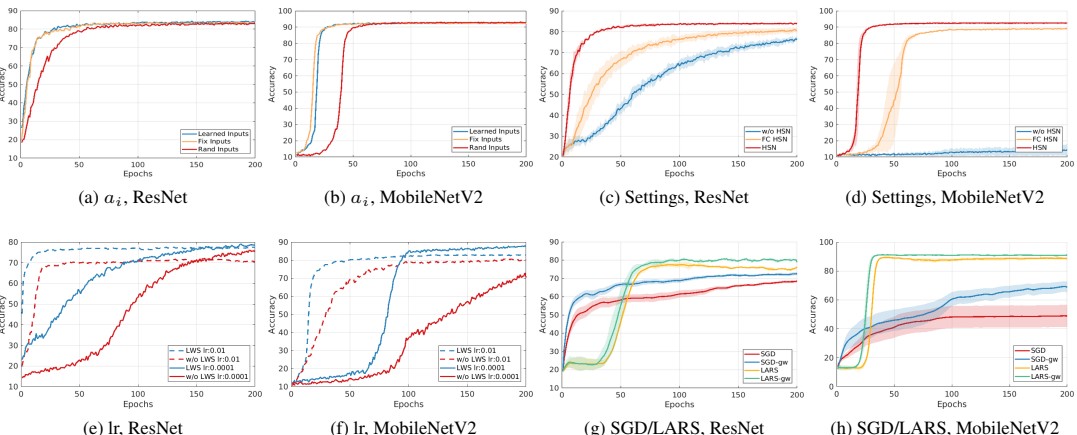

Figure 5: (a,b): Effect of different scheme for the inputs of HSN $a_i$. (c,d): Effect of different settings of HSN. (e,f): Effect of different learning rates with LWS. (g,h): Effect of different optimizer on with LWS. For plots in (c,d,g,h), shaded areas represents variance from 5 trials.

To answer the first question, we examine three different settings: learnable inputs, fixed inputs, and randomly generated inputs from the uniform distribution. From Fig. 5 (a,b), we observe that fixed inputs have similar performance to learned inputs, and both of them outperform random inputs. The idea of using fixed inputs is that we want to project the optimal sub-network to fixed vectors in the input space, which is generally simple (compared to learned inputs) and easy to train (compared to random inputs). The above results justify why we use fixed inputs.

To verify the effectiveness of different components of HSN, we use three different settings: vanilla HSN, HSN only with dense layers and gates (definition is given in Appendix). From Fig. 5 (c,d), it can be shown that HSN has the best performance, which again shows that we should not separately treat each channel or each layer.

In Fig. 5 (e,f), we plot training curves for different learning rates with or without LWS. It can be seen that LWS can lead to better performance, given different learning rates. Finally, in Fig 5 (g,h), we examine whether LWS is still useful given two additional optimizer: SGD and LARS (You et al., 2017). LARS applies layer-wise learning rates on overall gradients, which can be complementary to LWS. When applying LWS on these two methods, it still improves performance. SGD is not a good choice when the optimization involves discrete values, as suggested by the previous study (Alizadeh et al., 2019).

## 5    CONCLUSION

In this paper, we proposed a hyper-structure network for model compression to capture inter-channel and inter-layer relationships. An architecture vector can be generated from HSN to select a sub-network from the original model. At the same time, we evaluated this sub-network by using classification and resource losses. The HSN can be updated by the gradients from them. Moreover, we also identified the problem of FLOPs constraint (bias towards latter layers), which limits the final search space of HSN. To solve it, we further proposed layer-wise scaling to balance the gradients. With the aforementioned novel techniques, our method can achieve state-of-the-arts performance on ImageNet with four different architectures.

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

# A   VISUALIZATION OF PRUNED ARCHITECTURES

In Fig 6, we visualize the pruned architecture for ResNet-50 and MobileNetV2.

# B   BIAS OF FLOPS REGULARIZATION

We briefly discuss two types of FLOPs regularization used in our paper and trainable gate (TG) (Kim et al., 2020). First, we provide the specific definition of $T(v_i)$ (FLOPs of $i$th layer):

$$T(v_i) = K_i^2 \frac{\mathbf{1}^T v_{i-1}}{\mathcal{G}_l} \mathbf{1}^T v_i W_i H_i, \tag{7}$$

where $\mathcal{G}_i$ is the number of groups in a convolution layer, $K_i$ is the kernel size, $\mathbf{1}$ is a all one vector, and $\mathbf{1}^T v_i$ is the number of perversed channels in $i$th layer. With $T(v_i)$, $T(\mathbf{v}) = \sum_{i=1}^{L} T(v_i)$. In TG, they simply use mean square error (MSE) as the regularization term, and in their paper $\mathcal{R}_{\text{MSE}}(T(\mathbf{v}), pT_{\text{total}}) = (T(\mathbf{v}) - pT_{\text{total}})^2$. The gradients w.r.t $v_i$ is:

$$\frac{\partial \mathcal{R}_{\text{MSE}}}{\partial v_i} = 2(T(\mathbf{v}) - pT_{\text{total}}) \frac{\partial T(v_i)}{\partial v_i}, \tag{8}$$

For the regularization used in our method: $\mathcal{R}(T(\mathbf{v}), pT_{\text{total}}) = \log(|T(\mathbf{v}) - pT_{\text{total}}| + 1)$, the gradients w.r.t $v_i$ is:

$$\frac{\partial \mathcal{R}}{\partial v_i} = \frac{1}{|T(\mathbf{v}) - pT_{\text{total}}| + 1} \frac{T(\mathbf{v}) - pT_{\text{total}}}{|T(\mathbf{v}) - pT_{\text{total}}|} \frac{\partial T(v_i)}{\partial v_i}. \tag{9}$$

For both regularization functions, the ratio between the gradients w.r.t $v_i$ of two layers $k, j$ is $\frac{\partial T(v_k)}{\partial v_k} / \frac{\partial T(v_j)}{\partial v_j} = \frac{K_k^2 \frac{\mathbf{1}^T v_{k-1}}{\mathcal{G}_k} W_k H_k}{K_j^2 \frac{\mathbf{1}^T v_{j-1}}{\mathcal{G}_l} W_j H_j}$. Take ResNet-50 as an example, let $j, k$ be the middle layers of a bottleneck block, and we random initialize HSN. If $j$ is in the first block, and $k$ is in the last block, then $K_k = K_j = 3$, $W_j = H_j = 56$, $W_k = H_k = 7$, $\mathbf{1}^T v_{j-1} \approx 0.5 \times 64$ (due to random initialization), $\mathbf{1}^T v_{k-1} \approx 0.5 \times 512$, finally, $\frac{\partial T(v_k)}{\partial v_k} / \frac{\partial T(v_j)}{\partial v_j} \approx \frac{3 \times 3 \times 256 \times 7 \times 7}{3 \times 3 \times 32 \times 56 \times 56} \approx \frac{1}{8}$, which is not trivial.

When calculating the gradients w.r.t $\theta_i$, we have $\frac{\partial \mathcal{R}}{\partial \theta_i} = c_{\mathcal{R}} \frac{\partial T(v_i)}{\partial v_i} \frac{\partial v_i}{\partial \theta_i}$, all $\theta_i$ share the same $c_{\mathcal{R}}$ decided by the regularization function. Without loss of generality, we assume the magnitude of $\frac{\partial v_i}{\partial \theta_i}$ is similar given different layers. The assumption is based on the following derivation (to simplify derivation, we omit weight norm in dense layers):

$$\frac{\partial v_i}{\partial \theta_i} = \frac{\partial z_i}{\partial \theta_i},$$
$$= \frac{\partial z_i}{\partial o_i} \frac{\partial o_i}{\partial \theta_i},$$
$$= \frac{1}{\tau} \text{sigmoid}((o_i + g)/\tau)(1 - \text{sigmoid}((o_i + g)/\tau)) \frac{\partial o_i}{\partial \theta_i} \leq \frac{1}{4\tau} b_i^T.$$

where $\text{sigmoid}(x)(1 - \text{sigmoid}(x)) \leq \frac{1}{4}$, and $b_i$ is the input to $i$th dense layer, which is also the outputs of GRU. Since all $b_i$ have the same shape, and weights in GRU are normalized, we can assume all $b_i$ have similar magnitude. Since $\frac{1}{4\tau} b_i^T$ is a upper bound of $\frac{\partial v_i}{\partial \theta_i}$, similar assumptions can be made.

Following this assumption, the relative magnitude of gradients w.r.t $\theta_j$ and $\theta_k$ for layers $j, k$ can be roughly represented by $\frac{\partial T(v_k)}{\partial v_k} / \frac{\partial T(v_j)}{\partial v_j}$. After training for a while, the ratio might be smaller, however, it only indicates that early layers are more aggressively pruned. Thus, when applying FLOPs regularization, it penalizes early layers much heavier compared to latter layers.

One should also note that this is a general problem when using gradient based model compression methods with the FLOPs regularization. It's quite hard to circumvent calculating $\frac{\partial T(v_i)}{\partial v_i}$ as in TG (Kim et al., 2020) and our paper.

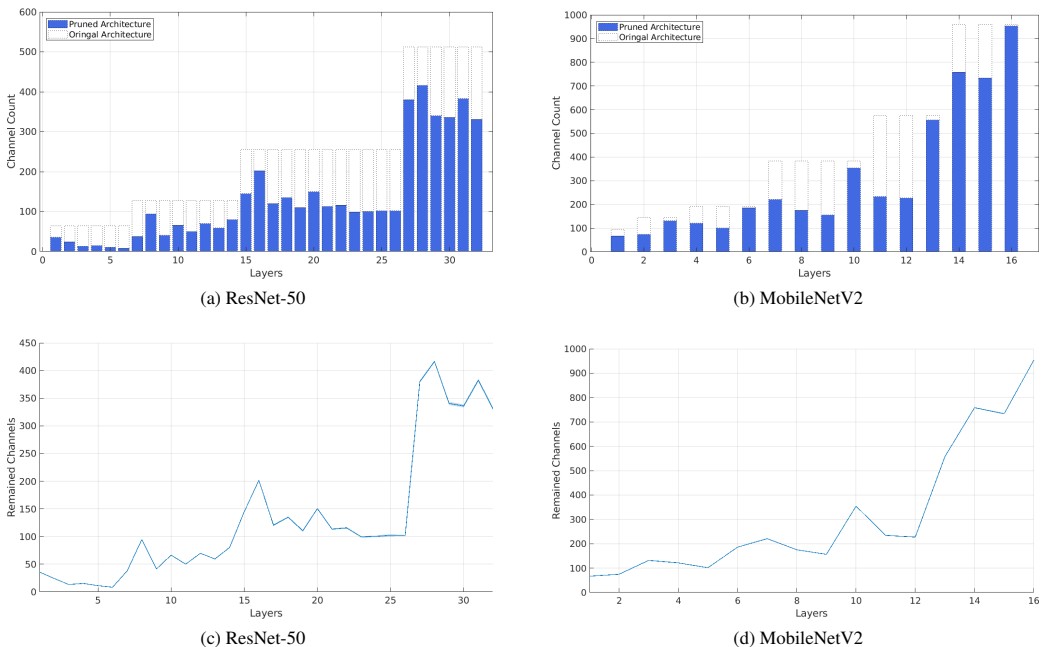

Figure 6: (a,b): visualization of pruned architectures for ResNet-50 and MobileNetV2. (c,d): mean and variance of 20 generated sub-networks for pruning.

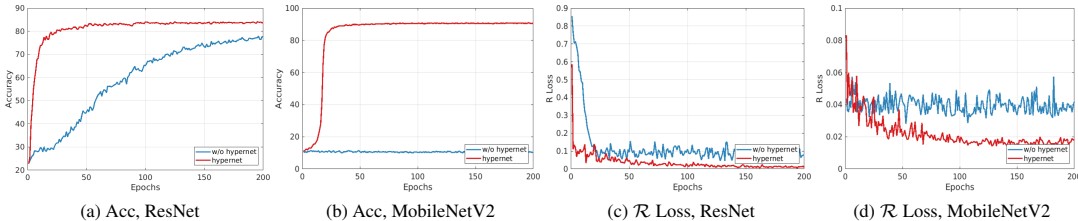

Figure 7: (a,b): Performance of sub-networks when using HSN or not using HSN (the setting in Eq. 11). (c,d): Regularization loss for the same settings.

## C    DETAILED SETUP OF HYPER-STRUCTURE NETWORK

In Tab. 3, we present the architecture of HSN. The forward calculation is:

$$b_i, h_i = \text{GRU}(a_i, h_{i-1})$$
$$o_i = \text{dense}_i(b_i) \tag{10}$$

where $h_i$ and $b_i$ are hidden states and outputs of GRU at step $i$, $o_i$ is the final output of HSN. GRU also requires hidden layer input at time-step 0 $h_0$. In the experiment, the $h_0$ is a all zero tensor. As mentioned in Tab. 3, the dimension of $a_i$ is 64. Since $a_i$ is a single input instead of a mini-batch, we cannot apply batchnorm. To make the training more stable, we use weight norm (Salimans & Kingma, 2016) on both GRU and dense layers.

Initially, we tried to use a huge dense layer (input size 64, output size $C_1 + C_2 + \cdots + C_L$) as HSN. However, we find that the huge dense layer is hard to optimize and also parameter heavy.

To verify the strength of the proposed HSN, we can instead use a simplified setting to prune neural networks, which is shown as follows:

$$\hat{z}_i = \text{sigmoid}((\hat{\theta}_i + g)/\tau),$$
$$\hat{v}_i = \text{round}(\hat{z}_i), \text{ and } \hat{v}_i \in \{0, 1\}^{C_i}, \tag{11}$$

| Inputs $a_i$, i=1,$\cdots$, L |
|:---:|
| GRU(64,128), WeightNorm, Relu |
| dense$_i$(128,$C_i$), WeightNorm, i=1, $\cdots$, L |
| Outputs $o_i$, i=1, $\cdots$, L |

Table 3: The structure of HSN used in our method.

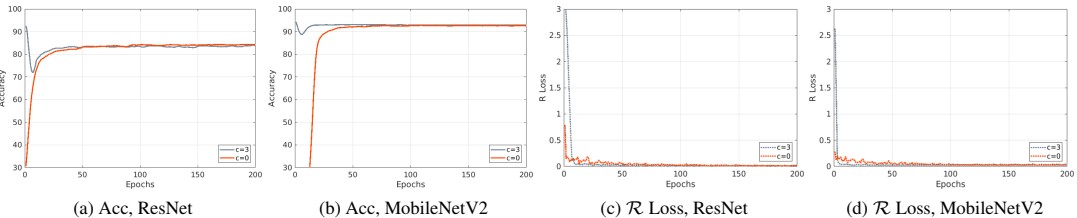

(a) Acc, ResNet      (b) Acc, MobileNetV2      (c) $\mathcal{R}$ Loss, ResNet      (d) $\mathcal{R}$ Loss, MobileNetV2

Figure 8: (a,b): Performance of sub-networks when training HSN given forward (c=0) and backward pruning (c=3). (c,d): Regularization loss of sub-networks when training HSN given forward (c=0) and backward pruning (c=3). All experiments are done on CIFAR-10.

where the architecture vector is parameterized by $\hat{\theta}_i$. Under this setting, the parameter for each channel does not have relationships. We use this setting to prune ResNet-56 and MobileNetV2 on CIFAR-10, the results are shown in Fig. 7. From the figure, we can see that the performance and convergence speed of using HSN is much better. Under high dimensional setting, like MobileNetV2, the simplified setting shown in Eq. 11 can not learn efficiently, which demonstrate that capturing inter-channel and inter-layer relationships are crucial for pruning deep neural networks.

## D    FORWARD AND BACKWARD PRUNING

Here, we refer forward pruning as start pruning from a random sub-network, and refer backward pruning as start pruning from the original large model. Many model compression methods use backward pruning. We also provide a simple way to extend our method to backward pruning. When we binarize the output of HSN, we can add a constant $c$:

$$
\begin{aligned}
z_i &= \text{sigmoid}((o_i + (g + c))/\tau), \\
v_i &= \text{round}(z_i), \text{ and } v_i \in \{0, 1\}^{C_i},
\end{aligned}
\tag{12}
$$

where $g \sim \text{Gumbel}(0, 1)$, and the Gumbel$(0, 1)$ distribution can be sampled using inverse transform sampling by drawing $u \sim \mathcal{U}(0, 1)$ and computing $g = -\log(-\log(u))$. When the constant $c$ is big enough, it will make $v_i$ become an all one vector, thus the sub-network produced by HSN will start from the original large CNN. If we set $c$ to 0, then it will start from a random sub-network. In Fig. 8, we show the results of forward and backward pruning. It can be seen that they can achieve similar sub-network performance, but the changes in regularization loss various dramatically.

## E    DERIVATIVE OF HYPER-GRADIENT WITH ADAM OPTIMIZER

The update rule of ADAM for $\theta_i$ is shown in Alg. 2, and it is:

$$
u(\theta_i^{t-1}, \alpha_i^t) = \theta_i^{t-1} - \eta \hat{m}_t / (\sqrt{\hat{n}_t} + \epsilon),
\tag{13}
$$

---

**Algorithm 2:** ADAM optimizer for $\theta_i$

---

**Input**: $\eta, \beta_1, \beta_2 \in [0, 1)$: learning rate and decay rate for ADAM.
Initialize $m_0, n_0, t = 0$
**Update rule at step** $t$:
$m_t = \beta_1 m_{t-1} + (1 - \beta_1)(\alpha_i^t \frac{\partial \mathcal{L}}{\partial \theta_i^{t-1}} + \lambda \frac{\partial \mathcal{R}}{\partial \theta_i^{t-1}})$
$n_t = \beta_2 n_{t-1} + (1 - \beta_2)(\alpha_i^t \frac{\partial \mathcal{L}}{\partial \theta_i^{t-1}} + \lambda \frac{\partial \mathcal{R}}{\partial \theta_i^{t-1}})^2$
$\hat{m}_t = m_t/(1 - \beta_1^t)$
$\hat{n}_t = n_t/(1 - \beta_2^t)$
$\theta_i^t = u(\theta_i^{t-1}, \alpha_i^t) = \theta_i^{t-1} - \eta \hat{m}_t/(\sqrt{\hat{n}_t} + \epsilon)$

---

Then the derivation of $\frac{\partial u(\theta_i^{t-1}, \alpha_i^t)}{\partial \alpha_i}$ is:

$$\frac{\partial u(\theta_i^{t-1}, \alpha_i^t)}{\partial \alpha_i} = -\eta \frac{\partial \big(\hat{m}_t/(\sqrt{\hat{n}_t} + \epsilon)\big)}{\partial \alpha_i}$$

$$= -\eta \frac{-\frac{\partial \sqrt{\hat{n}_t} + \epsilon}{\partial \alpha_i} \hat{m}_t + \frac{\partial \hat{m}_t}{\partial \alpha_i}(\sqrt{\hat{n}_t} + \epsilon)}{(\sqrt{\hat{n}_t} + \epsilon)^2}$$

$$= -\eta \big( \frac{\frac{\partial \hat{m}_t}{\partial \alpha_i}}{\sqrt{\hat{n}_t} + \epsilon} - \frac{\frac{\partial \hat{n}_t}{\partial \alpha_i} \hat{m}_t}{2\sqrt{\hat{n}_t}(\sqrt{\hat{n}_t} + \epsilon)^2} \big)$$

$$= -\eta \Big\{ \frac{(1 - \beta_1)\frac{\partial \mathcal{L}}{\partial \theta_i^{t-1}}}{(1 - \beta_1^t)(\sqrt{\hat{n}_t} + \epsilon)} - \frac{(1 - \beta_2)(\alpha_i^t(\frac{\partial \mathcal{L}}{\partial \theta_i^{t-1}})^2 + \lambda \frac{\partial \mathcal{L}}{\partial \theta_i^{t-1}} \frac{\partial \mathcal{R}}{\partial \theta_i^{t-1}})\hat{m}_t}{\sqrt{\hat{n}_t}(\sqrt{\hat{n}_t} + \epsilon)^2(1 - \beta_2^t)} \Big\},$$

where $\frac{\partial \hat{m}_t}{\partial \alpha_i} = \frac{(1-\beta_1)\frac{\partial \mathcal{L}}{\partial \theta_i^{t-1}}}{1-\beta_1^t}$ and $\frac{\partial \hat{n}_t}{\partial \alpha_i} = \frac{2(1-\beta_2)\big(\alpha_i^t(\frac{\partial \mathcal{L}}{\partial \theta_i^{t-1}})^2 + \lambda \frac{\partial \mathcal{L}}{\partial \theta_i^{t-1}} \frac{\partial \mathcal{R}}{\partial \theta_i^{t-1}}\big)}{1-\beta_2^t}$. Recall that when updating $\alpha_i^{t-1}$ to $\alpha_i^t$, we have to compute:

$$\alpha_i^t = \alpha_i^{t-1} - \beta \frac{\partial J(u(\theta_i^{t-2}, \alpha_i^{t-1}))}{\partial \alpha_i} = \alpha_i^{t-1} - \beta (\frac{\partial \mathcal{J}}{\partial \theta_i^{t-1}})^T \frac{\partial u(\theta_i^{t-2}, \alpha_i^{t-1})}{\partial \alpha_i}. \tag{14}$$

We need $\alpha_i^t$ when updating $\theta_i^{t-1}$ to $\theta_i^t$. Thus at each update step, it requires an extra copy of $\theta_i^{t-2}$ and parameters of ADAM to compute $\frac{\partial u(\theta_i^{t-2}, \alpha_i^{t-1})}{\partial \alpha_i}$. The cost of the extra storage is trivial.

## F    CHOICE OF $p$ GIVEN DIFFERENT DATASETS AND ARCHITECTURES.

| Dataset | CIFAR-10 | | ImageNet | | | |
|---|---|---|---|---|---|---|
| Architecture | ResNet-56 | MobileNetV2 | ResNet-34 | ResNet-50 | ResNet-101 | MobileNetV2 |
| p | 0.50 | 0.60 | 0.55 | 0.38 | 0.42 | 0.64 |

Table 4: Choice of $p$ for different models. $p$ is the **remained** FLOPs divided by the total FLOPs

In Tab. 4, we list the choices of $p$ for different models and datasets used in our experiments.

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
