# OpenReview forum: "Model Compression via Hyper-Structure Network"
_ICLR.cc/2021/Conference — Reject_

### Official Review · AnonReviewer1 · 2020-10-27

**Rating:** 6
**Confidence:** 5

**Review:**

## Summary
This paper proposes hyper-structure network for model compression (channel pruning). The idea is to have a hyper-network generate the *architecture* of the network to be pruned. To do so, the proposed approach use Gumbel softmax together with STE to get around the non-differentiability issue of such design. Additionally, the proposed approach dynamically adjust the gradient for each layer so that earlier doesn't get over-regularized due to the FLOPs regularizer. Empirical results are presented showing better performance compared to alternatives with ablation study on the proposed components (hyper-network and layer-wise scaling).

## Reasons for score
I'm leaning toward a rejection. I like the idea of both layer-wise scaling and using GRU to model inter-layer dependencies. However, I find them under-studied as the novel components of the paper. While this paper provides seemingly good performance compared to prior methods, they are not really apple-to-apple comparisons, which makes them relative weak signals. Moreover, some of the recent papers that are closely related to this submission are missing. I'm willing to raise my score if the weaknesses I listed below are properly addressed during the rebuttal period.

## Strengths

- The paper proposed a novel formulation towards the channel pruning problem. Specifically, the novelty lies in using GRU with the proposed layer-wise scaling.
- The results seem good compared to prior literature (with a caveat of having longer training iterations)
- Ablation of the proposed method ($\lambda$ and LWS)

## Weaknesses

- The novel aspect of the work, namely the layer-wise scaling, can be discussed in more detail. For example, why do we expect modifying the scaling of layer-wise gradient to make a better gradient than analytical gradient? In my understanding, analytical gradient gives you the steepest descent direction. In this case, why do we expect we can do better by tuning $\alpha_i$? It seems this is suggesting that we need layer-wise learning rate and such learning rates can be optimized via gradient descent by meta-gradient. It is not clear to me why such a formulation is specific to pruning. Can it benefit training vanilla network by setting $\lambda=0$? Overall, it is a bit mysterious to me why such a formulation *accelerates* training for equation (4), which is empirically shown in Figure 3 (c)(d). A more in-depth theoretical analysis would be very helpful. Without theoretical analysis, I'd be curious to see more settings empirically. If we sweep multiple learning rates, can LWS stop being better? If we use a different optimizer (say SGD), is LWS still better? Most importantly, if we use LARS optimizer [1], is LWS still better?

- Missing AutoML-based related methods that have strong performances [2-5]. Discussion with these prior methods are needed to better position the proposed method. Comparing with these methods, the proposed method is only comparable. Specifically, DMCP [4] has 47% reduction with top-1 of 76.2 for ResNet-50 and 30% reduction with top-1 of 72.2 for MobileNetV2. Putting these methods into the table and discussion is necessary.

- Comparison with AMC is limited to numbers from the previous paper. From the formulation of this paper, AutoML-based methods are highly relevant. It would be better if AMC is compared with this paper by using the same empirical setup. This is in fact doable as AMC has open source code. Without such a fair comparison, it is hard to understand what are the benefits of the proposed approach over AMC that solves the exact same problem. The paper has argued in the related work that policy gradient is noisy without really providing the quantitative evidences. The numbers from the paper is a really weak signal as AMC only fine-tunes for 30 epochs while this paper fine-tunes for 100 epochs for ImageNet.

- The other novel aspect of the paper is using GRU for designing the network. However, it is not clear if GRU is necessary and why it is a good design choice. The argument for GRU is cross-layer dependences. I'm wondering what the results would look like if we simply use FC for each layer independently. This can better motivate the so-called cross layer dependencies and better motivate the adoption of GRU.


[1] You, Yang, Igor Gitman, and Boris Ginsburg. "Large batch training of convolutional networks." arXiv preprint arXiv:1708.03888 (2017).

[2] Yu, Jiahui, and Thomas Huang. "AutoSlim: Towards One-Shot Architecture Search for Channel Numbers." arXiv preprint arXiv:1903.11728 (2019).

[3] Berman, Maxim, et al. "AOWS: Adaptive and optimal network width search with latency constraints." Proceedings of the IEEE/CVF Conference on Computer Vision and Pattern Recognition. 2020.

[4] Guo, Shaopeng, et al. "DMCP: Differentiable Markov Channel Pruning for Neural Networks." Proceedings of the IEEE/CVF Conference on Computer Vision and Pattern Recognition. 2020.

[5] Chin, Ting-Wu, et al. "Towards Efficient Model Compression via Learned Global Ranking." Proceedings of the IEEE/CVF Conference on Computer Vision and Pattern Recognition. 2020.


--------- Post rebuttal ---------------

I've read the rebuttal and I appreciate the additional efforts by the authors.

Specifically, the authors have addressed my concerns comparing LWS (the proposed method) and LARS. Additionally, the authors have addressed my concerns regarding LWS by conducting more experiments. With another detailed read, I figured LWS updates $\alpha$ in a lookahead fashion. Specifically, $\frac{\partial \mathcal{J}}{\partial \mu}$ in Eq. 6 essentially requires one to compute the loss after the gradient update is being made, which gives it the potential to outperform the analytical gradient.

Moreover, the authors have run additional experiments to demonstrate the usefulness of GRU, which makes the proposed method more convincing.

While the authors argued that it is not fair to compare to AutoSlim, AOWS, and DMCP, I disagree. They are all relevant and strong channel pruning methods and the authors should have cited them and discuss the main differences (can be used to prune a pre-trained model or not) in the related work as opposed to omit them entirely.

Overall, I find the paper interesting and it provides several novel aspects: GRU and LWS. Both are empirically verified to be useful in the channel pruning setting. However, the related work section can be further improved. As a result, I raised my score to weak accept.

---

> ### Author Response · Authors · 2020-11-23
> **Author response**
>
> Thank you for your constructive comments. We revised the paper and added additional experiments to address your questions.
>
> 1. In stochastic optimization, the stochastic gradient has variance; thus, the stochastic gradient is biased to the true full gradient (only unbiased in expectation). The search space of sub-networks is decided by the problem in Eq.4. The search space is not optimal itself, and thus, even though the gradients from this objective is unbiased, it still cannot achieve optimal results. Modifying the direction of gradients dynamically changes the search space, leading to better sub-networks even though its direction is different from analytical gradients.
>
>   Note that using LWS is not equal to setting a layer-wise learning rate, a layer-wise learning rate is applied on all gradients of a
>   layer. LWS only changes the scale of gradients from the classification loss, which also changes the direction of the final gradients.
>   The motivation is that the search space defined in Eq.4 may not be optimal. When lambda=0, using LWS cannot change the
>   direction of gradients, thus modifying it does not lead to significant changes.
>
>   LARS cannot be compared with our method directly, recall that the layer-wise learning rate in LARS modifies all gradients
>   together, and it also cannot change the direction of gradients. LWS can also be applied to LARS. In our revised paper, we plot
>   results for different learning rates and optimization methods in Fig. 5. From the figure, it’s obvious that LWS can improve both
>   SGD and LARS.
>
> 2. We updated Tab. 2 and added LeGR as a comparison method.
> For the rest methods, we believe it’s unfair to compare our method with them. All three methods (AutoSlim, AWOS, DMCP) change the training of the model from scratch. After modifying the training, the channel is implicitly ranked by its index, which vastly reduces the difficulty when searching sub-networks. On the other hand, our method can prune any pre-trained models without an assumption on the ranking of channels. In summary, their methods cannot prune pre-trained models used in our experiments. Our method considers both width and positions of channels. Their methods, on the other hand, only consider the width.
>
>   Other papers have already shown policy gradients are noisy. In Figure 9(a) of LeGR, they plot validation accuracy compared to
>   AMC by using DDPG. From that figure, you can see how noisy they are. The large variance of policy gradients has also been
>   shown in previous theoretical works [1, 2].
>
>   [1] Zhang, Kaiqing, et al. "Global convergence of policy gradient methods to (almost) locally optimal policies." arXiv preprint
>   arXiv:1906.08383 (2019).
>
>   [2] Shen, Zebang, et al. "Hessian aided policy gradient." International Conference on Machine Learning. 2019.
>
> 3. We carefully checked their source code but found that they currently do not support finetuning MobileNetV2 on ImageNet. In fact, in their original paper, they did not mention the detailed finetuning setup for MobileNetV2. They only said they finetune ResNet-50 (weight pruning instead of structural pruning) 30 epochs. Moreover, in their scripts, they finetune MobileNetV1 for 150 epochs using a cos-annealing learning rate scheduler. The finetuning of MobileNetV2 should be the same. The link to their finetuning script: https://github.com/mit-han-lab/amc/blob/master/scripts/finetune_mobilenet_0.5flops.sh. After checking their code, we also used a cos scheduler to finetune our model, and result in a better performance for MobileNetV2 (see updated Tab. 2).
>
> 4. We revised our paper and added extra results using a hypernet without GRU layers in Fig.5. Clearly, the connection of different layers is useful to search for a sub-network.

---

### Official Review · AnonReviewer4 · 2020-10-28
**#Official Review 4**

**Rating:** 4
**Confidence:** 5

**Review:**

This paper propose to utilize a hyper-structure network to generate the architecture of the main network, which can help to capture inter-channel and inter-layer relationships. The learnable layer-wise scaling factors are introduced to balance the gradients from different terms. BasisNet can be applied to any network architectures. MCH shows state-of-the-art ImageNet performance in pruning setting.

My main concern is about the novelty of the paper. The idea to use a hypernet to help pruning is not new. MetaPruning uses a hypernet for automatic channel pruning [1]. GaterNet use a gater network to generate pruning mask for each input image [2]. This paper use a hypernet to generate mask to prune the main network, which can be viewed as a weak version of GaterNet where the mask for each sample is the same.

In addition, Eq.5 involves 2 hyperparameters, i.e. $\lambda$ and $\alpha_i$. $\lambda$ is set by hand and $\alpha_i$ is learnable. The two hyperparameters may be deprecated, that is, you only need to preserve $\alpha_i$ and $\lambda$ can be merged into $\alpha_i$?

Overall, I think this paper below the bar of ICLR due to the above weakness.

[1] MetaPruning: Meta Learning for Automatic Neural Network Channel Pruning. ICCV 2019.
[2] You Look Twice: GaterNet for Dynamic Filter Selection in CNNs. CVPR 2019.

---

> ### Author Response · Authors · 2020-11-23
> **Author response**
>
> Thank you for your comments.
>
> 1. Thanks for pointing out these two papers. In related works, we have already compared our work with MetaPruning. MetaPruning generates weights of a CNN, and HSN generates a sub-network. The fundamental idea and implementation details are completely different.
>
>   Our method cannot “be viewed as a weak version of GaterNet”. First, the two papers consider completely different problems
>   (channel pruning vs. dynamic filter selection). There is no evidence showing which task is more or less difficult. Second, our
>   method aims to project the optimal sub-network to a sequence of vectors, which cannot be achieved by GaterNet. Third,
>   GaterNet does not explicitly consider the FLOPs constraint. On the contray, our method not only considers it but also includes
>   dynamic interactions between the classification loss and FLOPs constraint with LWS. As a result, we do not think GaterNet is
>   stronger than HSN or HSN is stronger than GaterNet. They are just not so relevant.
>
> 2. For the general cases (like SGD and LARS), it cannot be merged. With ADAM, it might be merged. Thus, we do not think it should be merged when describing our method. Moreover, we want to make Eq.4 and Eq.5 consistent.

---

### Official Review · AnonReviewer2 · 2020-10-28
**Some parameters need further investigations.**

**Rating:** 5
**Confidence:** 5

**Review:**

This work explores the model compression problem with a hyper-network to adaptively determine the preservation of inter-channel and inter-layers. To this end, they use a shared-GRU layer to explore the relations in consecutive layers and then FCs to generate the coefficient vectors which indicate the preserved rates within each layer. As a result, the inter-layer relation can be given from GRU and inter-channel relation can be given from FCs. In this paper, a learnable factor in each layer is also involved in the network optimization to better balance the classification performance and FLOPs regularization. Experimental results also validate the performance of the proposed method.

Overall, I think the paper is easy to follow and the derivation is also clear to understand. Nonetheless, there are still some problems that need to be further explored, given as follows:

1. This paper only investigates the channel pruning, but the title uses the "model compression". This might be not very specific since model compression serves as a general topic and consists of many approaches, such as channel pruning, quantization, knowledge distillation and tensor decomposition.

2. This paper also uses FCs to generate the coefficient vectors (ideally 0-1 code) to indicate the importance for each channel. However, this practice has been investigated in other channel pruning papers and is thus not new. Just to name a few,

[NeurIPS2019] AutoPrune - Automatic Network Pruning by Regularizing Auxiliary Parameters
[Pattern Recognition] AutoPruner: An End-to-End Trainable Filter Pruning Method for Efficient Deep Model Inference

3. In subsection 4.4, the experiments on Figure (a)-(b) reveal that “changing $\lambda$ does not have a large impact on the final performance of a sub-network, and our method is not sensitive to it”. I think the reason why the method is not sensitive toward $\lambda$ should be further analyzed. As in the appendix (E) shows, in Eq.(8), the update of $\alpha$ contains $\lambda$, so $\alpha$ can actually be reviewed as the adaptive variable, and for different $\lambda$ during the network training, the value of will gradually converge to the corresponding scale which considers $\lambda$.

4. I think there are some mistakes in analyzing the bias of FLOPs regularization. As stated in appendix (B), the relative magnitude of gradients w.r.t  $\theta_k$ and $\theta_j$ can be approximately estimated. However, since the ratio value of $v_k$ and $v_i$ is 8, the conclusion about "the gradient in the early layers are larger than that of the latter layers" does not hold. I wonder whether it is a typo here. Furthermore, can you explain why the assumption “the magnitude of  $\partial v_i \partial \theta_i$ is similar given different layers” holds herein?

5. In subsection 4.4, “In Fig.4(c,d) should be changed into “In Fig.3(c,d)” instead.

6. How to choose the hyper-parameter p?

---

> ### Author Response · Authors · 2020-11-23
> **Autor response**
>
> Thank you for your constructive comments. We revised the paper and added additional experiments to address your questions.
>
> 1. Thanks for your suggestion. We will change our paper title to ‘Channel Pruning via Hyper-Structural Net’ to be more informative in the final version. We didn’t change it in the current version because we do not want to bring confusion to other reviewers.
>
> 2. After carefully examining these two papers, we think that they are fundamentally different from our paper. In AutoPrune, they only introduce indicator functions for masks, which do not use any FC layers. AutoPruner uses FC layers to generate binary vectors given input feature maps, which is layer-wise independent, suggesting that they only consider sub-networks locally instead of globally. On the other hand, our method maps optimal sub-networks to a sequence of vectors building both inter-channel and inter-layer relationships by using a hyper-structural network. Additional experiments from Fig.5 in our revised paper also suggest that only using FC layers results in slower training and inferior final performance.
>
> 3. Thanks for your comments. With ADAM optimizer, the $\alpha_i$ will slowly adapt to lambda, which could be one reason why our method is not sensitive to lambda. We also modified our arguments in the revised paper. For other optimizers, the update of $\alpha_i$ may be irrelevant to lambda. Nonetheless, ADAM optimizer is the best optimizer for our method.
>
> 4. Indeed, this is a typo. It should be 1/8. We revised the supplementary materials and added detailed calculation steps. We also provided additional derivation to show why our assumption should approximately hold in supplementary matrials.
>
> 5. Thanks for your comments. We have corrected the typo.
>
> 6. It is quite straightforward to choose p. For a given CNN, we have its overall FLOPs: $T_{\text{all}}$, and the prunable FLOPs: $T_{\text{total}}$. Suppose we want to prune 50% FLOPs, we have p$T_{\text{total}}$ = 0.5$T_{\text{all}}$, and p = 0.5$T_{\text{all}}$/$T_{\text{total}}$.

---

### Official Review · AnonReviewer3 · 2020-11-02
**Review - Hyper-structure Network**

**Rating:** 5
**Confidence:** 3

**Review:**

This paper proposes a channel pruning method to compress and accelerate pre-trained CNNs. This paper proposes to use gated recurrent unit (GRU)-based hyper-structure networks (HSN) to generate channel importance, and Gumbel-rounding operations are subsequently applied to the importance values to generate stochastic masks to prune a CNN. The authors further introduced layer-wise trainable scaling with a modified update rule to accelerate convergence. While the tricks employed above are not original on their own, the combined method is still a novel amalgam, which learns inter-channel and inter-layer relationships in a unified package. Experiments on CIFAR-10 and ImageNet with ResNet and MobileNet-V2 showed that this method is competitive with existing state-of-the-art (SOTA).

### Advantages
* The combined method is novel, as it:
	* uses HSN to learn to prune without retraining models; and
	* learns inter-channel and inter-layer relationships in a unified HSN.
* Good results that are competitive with the SOTA.

### Issues
1. Section 3.2, “This process can be summarized as using Gumbel-Softmax distribution (Jang et al., 2016) to approximate Bernoulli distribution.” This is not true, as equation (2) clearly uses the “round” operation which gives discrete outputs, while the Gumbel-softmax approximation by Jang et al. is continuous with temperature annealing.
2. Section 3.2, the authors mentioned that $a_i$ is drawn from a uniform distribution but remains constant during training. What is the purpose of having only constant values that are not even hyperparameters as the inputs to the HSN? This seems unnecessary and introduces unwanted variance.
3. Section 3.4, “When generating the vector multiple times, most parts of vectors are the same, the different parts are trivial and do not have impacts on the final performance.” Why? More results are necessary to verify this.
4. The experiments are not run multiple times to give mean and standard deviation results.

### Other minor issues
* Section 2.1, “A straight forward way to”, should be “straightforward”.
* Figure 1 is not mentioned in text.
* In Section 4.4, “In Fig. 4 (c,d),”, should be Fig.3 (c,d).
* It is not apparent if layer-wise scaling was used for ImageNet models.

### Summary
While this method proposed in this paper is novel and the results are competitive, the reviewer believes that in its current state, this paper lacks additional experiments to address issues 2, 3 and 4 mentioned above.

---

> ### Author Response · Authors · 2020-11-23
> **Autor response**
>
> Thank you for your constructive comments. We revised the paper and added additional experiments to address your questions.
>
> 1. We revise the paper and change the description to ‘using ST Gumbel-Softmax with fixed temperature to approximate Bernoulli distribution.’ In Gumbel-Softmax, the temperature is annealed, but in Concrete distribution, the temperature is fixed. The idea of these two papers is very similar. In practice, the fixed temperature already achieves satisfactory performance.
>
> 2. In fact，there does not exist any biases in our experiment. It is infeasible to make them as hyperparameters due to their dimensionality (64 for each layer). The idea is to connect the best sub-network(s) with fixed representations in the input space, which generally reduces the variance of sub-networks and improves training speed. To show why fixed $a_i$ is a good choice, we provide additional results in our revised paper and adding two comparison baselines of $a_i$: (1) $a_i$ is randomly generated from uniform distribution during training (similar to the inputs of GAN). (2) $a_i$ is learned during training. From the figure, we can see that the performance of fixed $a_i$ is similar to learned $a_i$, and both of them outperform randomly generated $a_i$.
>
> 3. We visualized the mean and variance of 20 architectures generated by HSN for ResNet-50 and MobileNetV2 on ImageNet in supplementary materials. From these figures, we can see that the variance is negligible, which supports our arguments.
>
> 4. In the revised paper, we plot variance and mean by running experiments five times. The variance of HSN under ADAM training is quite low.
>
> 5. In our revised paper, we corrected the minor issues listed by the reviewer.

---

### Decision · Program_Chairs · 2021-01-07
**Final Decision**

**Decision:**

Reject

**Comment:**

This paper proposes a channel pruning method to compress and accelerate pre-trained CNNs.
The reviewers suggest further analysis of the experimental results to help explain the gains in performance, as well as point out some errors in the formulation. The paper is also found similar to meta-pruning method. The authors are encouraged to re-submit the paper after adding the analysis and improving the related work section.